# Implementing Lived Experience Workshops in Regional Areas of British Columbia to Enhance Clinicians’ Confidence in Spinal Cord Injury Care: An Evaluation

**DOI:** 10.3390/healthcare12070731

**Published:** 2024-03-27

**Authors:** Hannah Prins, Scott Donia, Shannon Rockall, James Hektner, Spring Hawes, James J. Laskin, John Chernesky, Vanessa K. Noonan

**Affiliations:** 1Praxis Spinal Cord Institute, Vancouver, BC V5Z 1M9, Canada; prinsh1@mcmaster.ca (H.P.); sdonia@praxisinstitute.org (S.D.); srockall@praxisinstitute.org (S.R.); jhektner@praxisinstitute.org (J.H.); shawes@praxisinstitute.org (S.H.); jlaskin@praxisinstitute.org (J.J.L.); jchernesky@praxisinstitute.org (J.C.); 2Faculty of Health Sciences, McMaster University, Hamilton, ON L8S 4L8, Canada; 3Department of Occupational Science and Occupational Therapy, University of British Columbia, Vancouver, BC V6T 1Z4, Canada

**Keywords:** spinal cord injury, medical education, quality in healthcare, healthcare delivery, regional healthcare, knowledge translation, community needs

## Abstract

In British Columbia (BC), there are challenges accessing specialized spinal cord injury care and resources. This paper evaluated the impact of spinal cord injury health educational workshops delivered in regional communities that were informed by persons with lived experience. A community survey was conducted with 44 persons with lived experience in a BC region to identify priority SCI health-related topics. Twenty-five topics were ranked from 1–14, with bowel and bladder management ranked 1 and 4, sexual health ranked 5, and pressure injuries ranked 7. Clinical perspectives on the priorities were collected from 102 clinicians in the BC region, who independently ranked 14 of these SCI topics and considered the former 4 topics to be lower clinical priority (ranked 11–14). These priorities informed a series of SCI clinical education workshops held at healthcare facilities in three regional cities. The goals were to improve clinicians’ knowledge and confidence levels when managing spinal cord injury health and to facilitate person-centred care. Positive feedback demonstrated that educational workshops supported by lived experience perspectives effectively enhanced the clinicians’ understanding of spinal cord injury and their priorities. Future plans include engaging more administrators as part of this initiative and conducting workshops in other regions of BC.

## 1. Introduction

Spinal cord injury (SCI) is a highly individualized, low-incidence injury that requires specialized knowledge, resources, and easy access to care. Despite being a rare injury, over 86,000 people in Canada are currently living with SCI [1], and persons with SCI are known to use health services more frequently than the general population across all medical care settings [2]. During the past 30 years, the global incidence, prevalence, and years lived with disability of SCI have increased [3], making it essential to provide effective and equitable management, prevention, resources and care for those with SCI.

Specialized healthcare services for SCI are commonly centralized in urban centres, creating barriers to accessing SCI health-related care for individuals living in regional communities. People with SCI who do not live near specialized centres, and those facing transportation barriers, have more healthcare needs that are unmet than do those living in urban centres [4]. With locally specialized care unavailable, many must travel long distances to urban areas to receive appropriate care, and may even relocate to urban areas [5]. This trend was recently observed in BC, where persons with SCI moved from regional to urban settings after injury for better care [6].

Across Canada and globally, this lack of adequate specialized regional healthcare stems from limited accessibility of services and insufficient healthcare providers to address gaps in the knowledge, literacy, treatment, and prevention of SCI [7,8]. In regional and rural parts of BC, occupational therapists and physical therapists proposed that rural contexts and access to services significantly shape definitions of health. They suggested that continued professional education is needed for clinicians to address gaps in knowledge and sustain optimal levels of care for individuals living with SCI in rural communities [9].

Intensive education and training workshops for specific or chronic health conditions are effective facilitators for improving clinical practices and changing attitudes, increasing knowledge, skills, and confidence levels among healthcare professionals [10,11,12,13]. An educational initiative on pressure injuries for critical care nurses in the United States demonstrated that a more comprehensive, subscale education model was needed to improve pressure injury prevention and management of pressure injuries [14]. Educational workshops prove instrumental in enhancing quality of care but can also help determine the next steps and identify areas for change. A Canadian initiative that engaged participants in workshops to establish research priorities in chronic pain emphasized the importance of knowledge transfer in project planning to improve the quality of life for those living with SCI [15].

For effective knowledge transfer, the educational content must address and support the current needs and priorities of individuals living with SCI in these regions. With the aging population, the topic of aging with SCI has become important and there is a need for evidence-based education in this area [16,17]. Other current key priorities for persons with SCI include upper extremity function, bladder and bowel function, sexual function, standing and walking, chronic pain, and stress management, as well as establishing a relationship with their clinician [18,19]. In providing this education, it has also been identified that there is a need to share experiences of what it is like to live with the secondary aspects of health beyond the physical, including peer support, mental health, and the social aspects essential to supporting individuals with SCI and their adjustment to life [16]. 

The objective of this paper is to describe a pilot quality improvement project to understand the priorities for persons with lived experience (PLEX) and the learning priorities for clinicians in the BC Interior, with the overall goal of facilitating person-centred care. In addition, the effectiveness of PLEX-informed educational workshops based on these priorities in increasing the knowledge and confidence of clinical team members was evaluated.

## 2. Materials and Methods

### 2.1. Persons with Lived Experience Participants and Priorities

The recruitment process for the PLEX Healthcare Priorities Survey was conducted through Accessible Okanagan’s private Facebook group. Accessible Okanagan promotes wheelchair-friendly discussions, activities, sports, and events throughout the greater Okanagan of BC. They currently have 731 members in their Facebook group. Members of the group were asked to participate in a survey aimed at gauging their perspectives on health-related topics. The survey asked participants to select and rank their healthcare priorities for health-related topics from 25 choices (see Appendix A Table A1). The data for PLEX Healthcare Priorities Survey were collected in August of 2021. Forty-four individuals completed surveys, but no identifying or demographic data were collected.

### 2.2. Clinician Participants and Priorities

Clinicians (e.g., physical therapists, occupational therapists, nurses, etc.) in three cities in the BC Interior were invited to participate in the workshops. Once individuals registered for the workshop, their confirmation included a request to participate in an anonymous Clinical Perspectives Survey. This 11-question survey asked several demographic questions (e.g., clinical role; level of care delivery) and questions about the experience, perceptions, and confidence levels related to working with individuals living with an SCI. Healthcare providers’ levels of confidence related to working with and treating SCI were also collected by ranking, from 1 to 5, their confidence in the subject matter. This clinical survey also asked participants to select and rank their priorities for the listed health-related topics for the educational workshops.

Surveys were returned by 102 of the 149 clinical team members who attended workshops in three BC Interior cities. Of the 102 participants, 42% were occupational therapists, 34% were physical therapists, 9% were rehabilitation assistants, 5% were in nursing roles, and 10% were other therapists/specialists or in non-clinical roles. The majority of these clinical team members worked in rehabilitation settings (*n* = 56), followed by acute/intensive care (*n* = 54), then community care (*n* = 30), long-term care (*n* = 10), and management/education (*n* = 10). Some of the respondents identified multiple roles. No personal identifying data were collected.

### 2.3. Educational Workshops—A Quality Improvement Project

The PLEX and Clinical Perspective Surveys on SCI-health-related topics that were identified as a priority informed a series of SCI clinical education workshops. The goals were to share information on these priorities and also to improve clinicians’ awareness, knowledge, and confidence levels when managing SCI health.

A series of 3 workshop sessions titled (1) SCI 101, (2) Pressure Injuries, and (3) PLEX Panel were conducted at each of the healthcare centres within the BC Interior from March 2022 to March 2023 (a total of 9 workshops). Each workshop was presented and moderated by an occupational therapist and a person with lived experience. SCI 101 was an introductory workshop covering the basics of SCI for clinicians. The Pressure Injury workshop focused on the prevention of pressure injuries and impactful personal stories, while the PLEX Panel explored the personal stories of PLEX from the acute to chronic stages. See Table 1 for a description of the 3 workshops. 

### 2.4. Post-Workshop Data Collection

After each workshop, the attendees were provided with a survey link and requested to complete anonymous, open-ended, 6-question surveys. These included questions gathering feedback about the quality of the workshop and workshop-specific information such as assessing clinical confidence regarding topics covered, most helpful discussion topics, perceptions of SCI healthcare needs, and barriers or challenges with delivering care (see Table 2).

### 2.5. Data Analysis

Given that the surveys were voluntary and anonymous, a descriptive analysis of the survey data was conducted. The relationships between the perspectives of those living with SCI and the healthcare providers for the priorities were determined by comparing the number of times each health topic was chosen by responders (PLEX or clinicians). The clinicians’ barriers and challenges to assessing, treating, and preventing pressure injuries were identified by examining their responses and counting how many times similar comments were made. The overall impact of the educational workshops and PLEX panel discussions and their impact on clinical practice was examined by two team members (HP and SD), by independently conducting a thematic analysis of participant feedback and reaching consensus on the key themes.

## 3. Results

### 3.1. Community Perspectives and Priorities

#### 3.1.1. Persons with Lived Experience Participants’ Community Priorities

PLEX community members (*n* = 44) completed the PLEX Perspectives Community Survey and identified the top 10 priority SCI health-related topics. See Appendix A Table A1 for the complete list of SCI health-related topics. The top three PLEX community priorities were aging with SCI, bowel function, and (tied) pain and inflammation and spasticity. 

#### 3.1.2. Clinical Team Members Priorities

Clinical team members (*n* = 102) completed a Clinical Perspectives Survey and identified the top 10 priority SCI health-related topics. Eleven of the health topics on the PLEX Survey were not part of the Clinician Survey (see NA listed in Table 3). The top three priorities for clinical team members were Spasticity, Upper Extremity Function, and Autonomic Dysreflexia. The Clinical Perspectives Survey results also showed that 81.2% of clinicians had experience working with individuals with SCI, 6.9% reported some experience, and 11.9% did not have any. The average time working clinically was four years.

#### 3.1.3. Comparing Community Rankings of Priority Spinal Cord Injury—Health-Related Topics

The PLEX Community Survey and Clinical Perspectives Survey gathered information on the SCI health-related priority topics for the 14 health areas that were part of both surveys. Participants were asked to select their main topics of interest, and these data were organized into rankings based on topics with the most selections (Table 3). The rankings of SCI health-related topics that were included in both surveys are compared in Figure 1. A comparison of the ranking of the PLEX priorities (*n* = 44) and those of the clinicians (*n* = 102) is included. 

The top 14 topics ranked by PLEX community members included bowel/bladder management (ranked 1 and 4), pain and inflammation (2), sexual health (5), and pressure injuries (7). These priorities differed from the clinical ones, especially for bowel/bladder management, sexual health, and pressure injuries, as they were ranked as the bottom four priorities (11–14).

### 3.2. Workshop Surveys and Feedback

Responses were open-ended and themes were identified for each question. 

#### 3.2.1. Spinal Cord Injury 101 Session Survey Results

The SCI 101 Survey asked survey participants (*n* = 29) what session topics they found helpful to give insights into the session’s impact and for future workshop planning. Results showed that clinicians most valued the following topics:1.Spasticity (15/29);2.Transitioning from rehab to community (14/29);3t.Bladder/bowel management (12/29);3t.Sexual health (12/29).

#### 3.2.2. Pressure Injury Workshop Survey Results

The Pressure Injury Workshop Survey asked participants (*n* = 39) what session messages they found most helpful. Across all three hospitals, the top four most valuable messages were identified:Hearing personal experiences of pressure injuries (33/39);Learning about quality of life and financial costs of PI (27/39);Impacts of healthcare system of pressure injuries (25/39);Prevention of pressure injuries (17/39).

Survey participants (*n* = 39) were asked to describe current challenges when preventing and treating pressure injuries. Challenges described included case variability, client involvement, lack of early identification of pressure injuries, equipment availability/maintenance, financial costs for clients, long healing/recovery process, inconsistent training and staff knowledge, offloading/repositioning techniques, quick onset time of pressure injuries, poor communication among staff, and staff shortages. These challenges were summarized into the following main categories:Better education for staff (24/39);Funding shortages (22/39);Awareness of the complicated/sensitive nature of pressure injuries (14/39);Client involvement * (12/39) (* Client participation in treatment plans is mainly out of the client’s control, making client involvement challenging to measure and improve).

The survey also asked participants (*n* = 39) to identify potential gaps in service implementation for individuals living with SCI in their region. The gaps were ranked in the following order:Access to specialized equipment/supplies (27/39);Workload limitation/time (26/39);Lack of specialized clinicians (22/39);Resources—where/how to find information (17/39).

This list of gaps in service implementation differs from the everyday challenges related to pressure injuries above, which focus on issues in clinicians’ roles, while gaps in service implementation address issues at the institutional/administrative level.

#### 3.2.3. Persons with Lived Experience Panel Session Survey Results

The PLEX Panel Survey asked survey participants (*n* = 31) what session topics they found helpful to assess the impact of, for future workshop planning. Results showed that clinicians most valued the following topics:1t.Transitioning from rehab to community (29/31);1t.Challenges with the healthcare system (29/31);3.Bladder/bowel management (24/31);4.Psychosocial adjustment to SCI (18/31).

Survey participants (*n* = 31) were asked how the workshop would impact their future clinical practice. The most common responses were grouped into the following top four themes: Increased knowledge and confidence (15/31);Better ability to provide client-centred care (10/31);Mindfulness of language used when speaking with PLEX (8/31);Expanding treatment goals (3/31).

The survey also collected clinical opinions (*n* = 31) on what they identified as potential gaps in service implementation for individuals living with SCI in their region. The gaps were ranked in the following order:1.Resources—where/how to find information (20/31);2.Lack of specialized clinicians (19/31);3t.Access to specialized equipment/supplies (16/31);3t.Workload limitation/time (16/31).

#### 3.2.4. Workshop Participants Feedback

In each Post-Workshop Survey, there was space for survey participants to include feedback and comments regarding the workshop and future recommendations. An thematic analysis was conducted for 33 quotes of the 99 surveys collected and summarized into major themes (see Table 4). 

## 4. Discussion

The purpose of this project was to improve the quality of care provided to PLEX from clinicians within regional areas of the BC Interior. The workshop information was chosen through PLEX-informed consultation and to improve care within these regional areas. Overall, there was a difference in the priorities from the perspectives of clinicians and the PLEX, and this provided an opportunity to create a shared understanding by engaging people with lived experience in both the planning and the delivery of the workshops. 

### 4.1. Comparing Experiences of Persons with Lived Experience with Those with Clinical Perspectives and Priorities

The PLEX Community Survey identified SCI health-related topics including aging with SCI, bladder and bowel function, pain and inflammation, spasticity, sexual health, and upper limb function as some of their top priorities. This list of top priorities is consistent with the literature, which has reported aging with SCI, upper extremity function, bladder and bowel function, sexual function, standing and walking, and chronic pain as the top priorities for PLEX [16,17,18,19,20,21,22]. 

In comparing the priorities of the PLEX community to the clinical priorities, there was a difference in the ranking of the priority of SCI health-related topics. As mentioned previously, the topics of bowel/bladder management, pain and inflammation, sexual health, and pressure injuries were of interest to the PLEX community but were not ranked as high by clinical team members. The topic of bowel and bladder management had the most prominent contrast as it was ranked 1st and 4th in terms of PLEX priority interest but 13th and 14th in terms of clinical priority interest. This difference in ranking is likely due to the mix of clinicians who attended the workshop, who were primarily occupational therapists (42%) and physical therapists (34%), and there were very few nurses (5%). However, individuals with SCI describe how bowel care needs do not fit with the pace and care delivery processes of acute hospitals, leaving many unmet needs in this area, so there is a need to ensure all SCI clinicians are aware of these health topics [23]. 

Differences in PLEX and clinical priorities are not novel observations, as there is prior evidence of disparities between PLEX’s and healthcare professionals’ perspectives. A study examining patient and clinician perceptions of social determinants of health for chronic diseases found that the prioritization of social needs by patients and clinicians was misaligned, and there was a lack of systematization of information flow between the two groups [24]. To provide person-centred care, it is important to address these discrepancies in SCI priority health-related topics by incorporating the perspectives of people with lived experience. In a study that used research priority-setting partnerships with patients and clinicians, investigators found inconsistencies between the treatments patients and clinicians wished to see and the treatments being evaluated by researchers [25]. Investigators discussed that some research questions important to patients and clinicians may never occur to researchers [25]. This supports the need to educate and align priorities between PLEX and clinical teams and between PLEX communities and researchers who establish priorities for healthcare services. This is especially impactful in the context of SCI as the research community values partnerships with clinicians to promote the “bench-to-bedside and back again” framework. Still, more effective partnerships with PLEX communities are required for the complete cycle of research and dissemination [26]. 

Furthermore, the clinicians who participated in these workshops had several years of experience working with individuals with SCI. The clinical survey results showed that 81.2% of clinicians had experience working with individuals with SCI, with an average duration of experience of 4 years. This highlights that despite clinicians having direct interactions and experience with SCI care, there is still a need to provide a broad range of education to ensure care is aligned with the priorities of the PLEX community. This shared learning will help fill these gaps in knowledge, develop a better awareness of community needs and priorities, and strive for better health outcomes for those living with SCI. 

### 4.2. Educational Workshop Feedback

Throughout the data analysis of the Pressure Injury Workshop and PLEX Panel Surveys, a predominant theme was the need for better staff education. This need for staff education, increasing awareness of resources and information, and addressing the lack of specialized clinicians can be improved with educational workshops within the BC Interior. Overall, the feedback from the workshops demonstrated the need for continued clinical education in hospital settings. Educational workshops like the ones conducted in the BC region that incorporated those with lived experience have the potential to impact clinical practice. Clinicians who attended the PLEX Panel with participants mentioned that the session equipped them with increased knowledge and confidence, a better ability to provide person-centred care, and increased mindfulness of language used when interacting with PLEX. 

Since educational initiatives are known to improve care and clinical expertise effectively, some next steps might include evaluating which method of education is most effective. When comparing this in-person, small-scale educational intervention method to others, the future models for continuing professional development and educational interventions are primarily described as e-learning and online. As the SCI workshops described in this evaluation initially began online and transitioned into an in-person model, it would be beneficial to investigate other workshop methods and compare them to those implemented in the BC Interior, especially since online measures could help expand workshops to other regional/rural communities. Some studies suggest that computer-aided learning is just as effective as conventional teaching methods for improving knowledge for healthcare students and educators [27] and would support using videoconferencing as an acceptable and effective method of delivering education for specialized conditions [28].

In contrast, another educational series on electrical stimulation therapy for pressure injuries found that online education effectively increased healthcare providers’ knowledge of the treatment. However, hands-on workshops were required to change certain attitudes for practice change [29]. A specific study with occupational therapists and the “Do-Live-Well” framework showed no difference in knowledge acquisition between in-person and online groups. Still, participants had greater satisfaction with in-person workshops [30]. Other research would suggest that “digital education was more effective than no intervention, while blended learning was superior to exclusive digital education” [31]. A technology-enabled knowledge translation framework for continuing medical education online has shown many benefits in accelerating the incorporation of the latest health evidence into routine practice and improving healthcare knowledge through interactive experiences [32]. These frameworks could serve as potential structures for future workshop series, but more research is required to discover the most effective method of educational interventions for SCI healthcare providers. 

Lastly, what makes these workshops unique from other educational initiatives are the PLEX perspectives shared and the leadership/facilitation of the workshops by a PLEX community member and clinician. The PLEX perspectives shared during workshops were especially beneficial to workshop participants as it was the overarching theme from feedback commentary and initiated deeper understandings of SCI-related conditions, priorities, and implications. A key focus of the Praxis Spinal Cord Institute and its programs is PLEX engagement and its active role in decision-making, setting priorities, conducting research, and translating research knowledge for the benefit of the intended users [33]. When considering the results of this evaluation and approaching the following steps and goal setting, there is a proven benefit to the incorporation of PLEX perspectives when educating and improving clinical practice [34]. Especially when investigating the remaining gaps in service implementation identified by this paper, it will be important to maintain and continue developing a person-centred approach hen looking into the next steps to address challenges.

### 4.3. Next Steps and Project Recommendations 

To advance the sharing of knowledge and resources among the three cities in the BC Interior, a meeting was held called BC Interior SCI Health Summit: Bridging Education for Lasting Impact Kelowna, BC, on 25 September 2023. This Summit brought together many community SCI stakeholders to learn, share experiences, and build connections to support SCI health within the BC Interior. As a direct result of this, a BC SCI network is being created to continue engagement in the Interior and focus on improving SCI care, accessibility, and communication. 

More specific recommendations for future workshop topics include bowel and bladder management due to the low clinical awareness and need for education in this area. One report investigated how SCI clinicians, researchers, government, and private funding organizations could share knowledge and examine emerging approaches to bowel and bladder dysfunction [35]. 

Another recommendation from the Pressure Injury Workshop was the poor communication among staff when handling pressure injuries. Managing these injuries requires an inter-disciplinary team, and tasks may vary between disciplines among regional communities. This makes effective communication between roles and departments especially important for the best quality of care. In the context of holding workshops to improve clinician skills, a similar educational workshop could be held to improve this challenge. An example of this communication training workshop was used in a paediatric intensive care unit and showed remarkable improvements in inter-specialty communications between clinicians in this area [36].

In order to address similar issues in regional communities lacking in SCI facilities and clinician knowledge, a similar approach to that demonstrated within this paper could be replicated. Through PLEX driven and informed educational workshops, more clinicians can become educated and more confident in working with PLEX and thus improve SCI care. Creating champions of change who can be trained to train others in regional communities could play a meaningful role in addressing some of the knowledge gaps. 

### 4.4. Limitations 

The themes and findings expressed in this paper represent results from a quality improvement project. It is important to note that the priorities were generated by a small team with the goal of identifying future workshop topics. Responses were limited to the choices given. The survey was not generative, and may not reflect the entire range of health priorities for people with SCI.

Community and workshop surveys were restrictive to three cities in the BC Interior and the surveys should be administered in other cities to determine if they are representative of the broader SCI community. Demographic information of workshop attendees that could potentially impact results was either not collected or undisclosed (e.g., race, sex, gender, age, occupation, familiarity with SCI, and number of times visited a workshop). The demographics of people with SCI who responded to the survey and/or attended the workshop may not represent the PLEX community throughout Canada. Members and associates of Accessible Okanagan are largely white, male, middle-aged, and relatively affluent. Many, if not most, have access to personal transportation, appropriate housing, recreational opportunities, and employment. In addition, many, if not most, are in relationships and/or have supportive social groups and/or families. Every person with lived experience who responded to the survey has access to the internet and technology and a level of comfort in navigating and voicing their opinions. Future surveys should focus on a more representative population from a larger area and with more diverse demographics to truly reflect the regional population. Demographic data should also be collected in future studies on PLEX priorities as those with different injury levels, years post-injury, ages, etc., may prioritize different topics. 

A comparison of the clinician pre and post workshop data were not included in this paper due to the discrepancy in pre and post numbers and the lack of a study identifier on the surveys. Given this current project was a quality improvement project it was not possible to collect this detailed data. Future research studies, however should be conducted to obtain more detailed data and link the pre and post survey results to facilitate more robust analyses. 

Finally, as mentioned previously, it is also possible that the roles of the clinical perspectives may have impacted the differences in priorities. The majority of respondents were physical and occupational therapists and may not have recognized the importance of topics such as the bowel and bladder as these are typically not part of their clinical practice. Some survey-specific limitations include the low response rates for the workshop surveys (an average response rate of 38%) and missing survey data for some questions. 

## 5. Conclusions

Overall, differences were observed between the priorities for the PLEX and clinicians for education. This project demonstrated the need and value of educational workshops presented by a clinical and PLEX team. By bringing these two perspectives together in the delivery of SCI education, there was an opportunity to broaden clinician knowledge and improve clinician confidence, which will help support the delivery of person-centred SCI care. Future work should continue to fill gaps in knowledge identified by the clinicians, and engage administrators to assist clinicians implement the newly acquired knowledge and resources.

## Figures and Tables

**Figure 1 healthcare-12-00731-f001:**
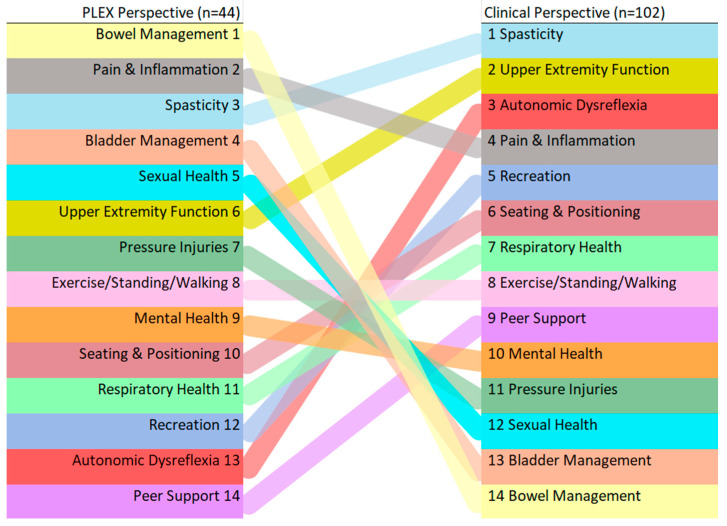
Ranking SCI topics of interest from PLEX and clinical perspectives.

**Table 1 healthcare-12-00731-t001:** SCI Workshop Descriptions.

Workshop Title	Description
SCI 101	This session was a high-level presentation geared for clinicians to enhance their knowledge of SCI, specific needs, and a variety of SCI topics to improve confidence when working with people with SCI. Information shared and issues discussed include the following: Canadian SCI Registry, best practices, common terminology for types/levels of injury, statistics, Canadian SCI Care pathway, effects of SCI on body systems beyond motor paralysis, International Standards for the Neurological Classification of SCI (ISNCSCI) Assessment overview, secondary health conditions, pressure injuries, neuropathic pain, autonomic dysreflexia, bladder/bowel function, respiratory health, sexual health, clinical care considerations, psychosocial adjustments to SCI, peer support, adaptive travel considerations, recreation opportunities, and sharing of resources.
Pressure Injuries	This session was focused on pressure injuries and was supported by PLEX personal stories and a presentation for SCI clinicians to enhance their knowledge of pressure injuries related to SCI and prevention techniques. Information shared included defining pressure injuries, stages, understanding their significance, statistics, costs of prevention, prevention methods/equipment, and many other resources. There was also time for Q&A discussions and PLEX sharing the personal impacts and their experiences related to pressure injuries.
PLEX Panel	A panel of 4–5 individuals with lived experience of SCI shared their journeys from the early days post-injury to transitioning to living in the community. Discussion topics varied from who was sharing and the questions asked by attendees and clinicians. PLEX panels also often included both men and women with a variety of injury types, levels of injury, paraplegics and tetraplegics, as well as new and experienced PLEX (from 6 months to 40 years post-SCI).

**Table 2 healthcare-12-00731-t002:** Questions for Participants After Attending the Workshops.

Question Number	Description
1	How familiar and confident would you consider yourself with the following topics related to SCI: autonomic dysreflexia, bladder/bowel management, pain management, pressure injury, psychosocial, recreational/peer opportunities, respiratory health, sexual health, spasticity? (note: each SCI topics are rated)
2	How useful did you find the session?
3	What messages from the discussion did you find helpful?
4	How will this session influence your future clinical practice?
5	As a clinician, what would you identify as potential gaps in service implementation for individuals living with SCI in your region?
6	For future workshops, what SCI-related topics would you like to learn more about, and which would be of most interest to your clinical practice?

**Table 3 healthcare-12-00731-t003:** Comparing SCI Topics of Interest between PLEX and Clinicians.

SCI Topic	PLEX Perspective (*n* = 44)		Clinical Perspective (*n* = 102)	
	N (%)	Rank	N (%)	Rank
Activity-Based Therapy	5 (11.4)	t-14	NA	
Aging With SCI	20 (45.5)	1	NA	
Autonomic Dysreflexia	2 (4.5)	23	61 (59.8)	3
Bladder Function/Management	11 (25.0)	t-5	37 (36.3)	13
Bowel Function/Management	13 (29.5)	2	36 (35.3)	14
Cardiovascular Health	6 (13.6)	t-9	NA	
Diabetes/Diet	6 (13.6)	t-9	NA	
Epidural and Transcutaneous Spinal Cord Stimulation	6 (13.6)	t-9	NA	
Equipment	11 (25.0)	t-5	NA	
Exercise, Standing, and Walking	5 (11.4)	t-14	54 (52.9)	8
Mental Health	5 (11.4)	t-14	45 (44.1)	10
Nerve and Tendon Transfer	5 (11.4)	t-14	NA	
Pain and Inflammation	12 (27.3)	t-3	60 (58.8)	4
Peer Support/Mentorship	1 (2.3)	t-24	48 (47.1)	9
Pressure Injuries	6 (13.6)	t-9	39 (38.2)	11
Recreational Options	3 (6.8)	t-20	59 (57.8)	5
Research Initiatives	NA		50 (49.0)	
Respiratory Health	3 (6.8)	t-20	56 (54.9)	t-6
Seating and Positioning (Wheelchair Mobility)	4 (9.1)	19	56 (54.9)	t-6
Sexual Health	10 (22.7)	7	39 (38.2)	t-11
Social Assistance Programs	5 (11.4)	t-14	NA	
Spasticity	12 (27.3)	t-3	78 (76.5)	1
Sport Performance/Paralympics	3 (6.8)	t-20	NA	
Upper Extremity Function	9 (20.5)	8	70 (68.6)	2
Vehicle Modifications and Transportation	1 (2.3)	t-24	NA	
Wheelchair Accessible Housing	6 (13.6)	t-9	NA	

NA: Not applicable, since these areas were not included in the clinician survey. t indicates a tie for the priority area.

**Table 4 healthcare-12-00731-t004:** Thematic Analysis of Workshop Feedback and Comments.

Main Theme (*n* = Frequency)	Quote Examples/Excerpts
Benefits of lived experience (*n* = 14)	“The personal experience was especially eye-opening.” “Hearing real stories. Sometimes, it is easier to remember important things if they are delivered via experience. I had no idea that bowel care/continence is so much of a big deal, if not hearing it first-hand.”
Learning about the secondary impacts of SCI (*n* = 7)	“It was also great to hear about the other aspects of what people with SCI deal with on a daily basis that we may not necessarily always think about.” “It makes us consider what other things we need to consider prior to discharge.” “So very moving. It’s easy to forget the ‘unseen’ injury that happens to the loved ones of our patients. This really helped to remind me of that, and what an incredible job the informal caregivers do in so many patients lives.”
Needs for resources (*n* = 7)	“How do I quickly and easily access expert advice when I have a patient with SCI.” “More funding options for equipment.”
Recommendations for future workshops (*n* = 5)	“Really enjoyed the session and wanted it to go on longer. Could be 2 or 2.5 h next time.” “It would be nice to write down a couple of the questions there were unable to be answered and address them on following sessions for 5–10 min.” “I wish I could have had a “Where are they now.”
Education is key (*n* = 5)	“How do we help educate and positively impact clients who have limited funding and decreased engagement in their wellness goals.” “Information given was actionable as well as informative.”
Require more clinical information (*n* = 3)	“I would just encourage the panel to keep it more clinical focused as best they can.”

## Data Availability

The original contributions presented in the study are included in the article/Appendix A, and further inquiries can be directed to the corresponding author.

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
