# Peer review of "Implementing Lived Experience Workshops in Regional Areas of British Columbia to Enhance Clinicians’ Confidence in Spinal Cord Injury Care: An Evaluation"

_healthcare, 2024, doi:10.3390/healthcare12070731_

Round 1
Reviewer 1 Report
Comments and Suggestions for Authors
In this manuscript, Prins et al. report and discuss the results of a survey of people with lived experience of spinal cord injury (SCI) and the results of clinician education workshops for SCI in regional areas of British Columbia. The topic of this manuscript is vitally important as there has often been great misalignment between researcher/clinician priorities and the issues that often have the greatest impact in the lives of individuals with SCI. Furthermore, individuals with SCI in regional areas often have less access to adequate equipment and care, requiring further education for clinicians in those regions. This manuscript contains interesting qualitative insights into how SCI care and clinical education can be improved. However, I have multiple significant methodological concerns with the manuscript that need to be addressed by the authors as discussed below.
Major Comments
· Abstract: “However, more interventions, clinical training sessions, and resources are required to address community needs and priorities.” This is a vague and general conclusion sentence for the abstract. It is somewhat obvious that more interventions, training, and resources would be beneficial. However, as stated in the introduction, regional and rural communities will often lack this type of infrastructure. A concrete statement on how the findings of this manuscript could inform on how to improve SCI care in regional areas that are unlikely to have access to increased resources would increase the relevance of this manuscript.
· There were 44 responses to the PLEX survey that was sent out to the Accessible Okanagan Facebook group. Is this group solely populated by individuals with SCI or does this group also contain caregivers and family members of individuals with SCI? As there was no demographic data collected, it is possible that the responses could be obtained from individuals without SCI. Additionally, it is possible that a Facebook group could contain individuals with SCI who are no longer in the Okanagan area, but remained in the group. Were there any additional components of the survey to ensure that the respondents fit the necessary criteria to be considered a PLEX in regional British Columbia?
· Why wasn’t demographic data collected during the PLEX survey? Inclusion of this information is critical to contextualize the results. For example, if the population sample was weighted towards lower vertebral injuries, upper extremity function may not be as large of a concern, and not listed among topics of interest.
· The methods indicate that the clinicians were asked to rank their confidence from 1-5 in the subject matter. Why wasn’t this data included in the manuscript? Were the clinicians polled again following the workshop? This would add significant quantification to suggest that the workshops could have a beneficial effect on clinician knowledge of SCI.
· Why were different topics asked on the PLEX survey and on the clinical perspective survey? As one of the main outcomes of this manuscript is a comparison between priorities of the PLEX and clinician communities, it would seem crucial to ask both groups to rank the same topics. For example, “Aging with SCI” was the highest ranked SCI topic from the PLEX community, but was not even asked of the clinical participants? This seems especially concerning as, even though Aging with SCI was the highest rank, to adequately compare across groups, the abstract, Figure 1, and discussion all indicate that bowel management was the highest ranked PLEX topic of interest, which is misleading.
· For Table 3, how many topics of interest were the participants allowed to choose/was there a difference in how this question was worded? The second highest scoring response for the PLEX survey was 29.5%, which is lower than any category for the clinical perspective survey. What could have caused this discrepancy?
· Were the priorities identified by clinicians what they wanted to learn more about or what they saw as priorities for the SCI community? Or was this left open to the interpretation of the respondents?
· For the different subsections in Section 3.2, the ranked categories vary in number from 2 to 7. Did these categories include all identified ranked responses? Otherwise, how were the number of ranked topics determined? It would be more objective to include a similar number across all categories or establish a threshold of responses (i.e., “Staffing shortages” was only reported by one participant, but is listed).
· The introduction goes into detail discussing the issues with addressing issues for the SCI population in regional/rural communities, and the workshops here address some of those issues in British Columbia. However, the discussion would be strengthened if these concepts were broadened to include a discussion about how similar approaches could be used in other regional/rural areas. This is especially important for areas that may not be near urban areas with relevant SCI equipment and facilities, or be in a province/area such as BC with significant SCI infrastructure.
Minor Comments
· Abstract, Line 11: “there are limited resources and challenges”. Maybe switch the order of “challenges” and “limited resources”? Currently, the “limited” could extend across the “and” to be read as “limited resources and limited challenges”.
· Abstract, Line 13: This should be “evaluates” as “This paper” is the singular noun.
· Abstract, Lines 17-19: “Clinical perspectives were collected from 102 clinicians in the BC Interior, who independently ranked 14 of these SCI topics but considered them to be a lower clinical priority (11-14).” This sentence is confusing as “them” is not defined. The lower ranked priorities should be stated or more concretely connected to the previous sentence.
· Abstract, Lines 21-24: “Positive workshop feedback surveys demonstrated that educational interventions supported by lived experience perspectives (42%) effectively enhance SCI health services and align clinician and PLEX perspectives.” What does the 42% here refer to? It is unclear based on how this is written, and 42% is not used again in the body of the text.
· The 8th reference is to a qualitative study of therapists and individuals with spinal cord injury in South Africa, but the sentence starts with “Across Canada…”. This is a reasonable reference for this manuscript, but seems to be placed in the wrong context.
· Missing period after “greater Okanagan of BC”, Line 84.
· What were the three cities invited to the BC Interior workshop? This information seems important to determine the regional/rural populations served by the clinicians.
· Missing end parentheses after “see NA listed in Table 3”, Line 159.
· The data shown in Section 3.2 is from the open-ended surveys discussed in Section 2.4? Therefore, the quantification for each category (e.g., spasticity, sexual health) was based on qualitative review? If so, this should be emphasized in the results section that these responses were not based on selections, but based on analyzing user reports.
· “transitioning” on line 185 is the only non-capitalized category.
· Line 236: “Resources – where/how to find information (2/31)” This is listed as the highest ranked gap in service ahead of another with 19/31, so this seems to be a typo?
· Line 250: “regional areas of the BC.” Is the “the” here correct?
· Line 252: missing comma after “Overall”
· Lines 258-261: “This list of top priorities is consistent with the literature, which has reported aging with SCI, upper extremity function, bladder and bowel function, sexual function, standing and walking, and chronic pain [16-22].” This doesn’t seem to be a completed expression. “has reported… as top priorities”?
Reviewer 2 Report
Comments and Suggestions for Authors
The article "Implementing Workshops in Regional Areas of British Columbia to Enhance Clinicians’ Confidence in Spinal Cord Injury Care: An Evaluation" delves into the crucial topic of enhancing clinicians' proficiency in spinal cord injury (SCI) care. The study reveals significant insights into the disparities between patient perspective and the perspectives of clinicians regarding SCI care. A notable finding highlighted in the article is the evident disparity between the perspectives of PLEX and clinicians. Despite a considerable portion of the clinical team surveyed possessing several years of experience in working with individuals with SCI, there remains a notable misalignment in priorities.
The findings emphasize the urgency for bridging the gap in knowledge and understanding between clinicians and the SCI community. The dissonance observed underscores the importance of fostering better awareness of community needs and priorities among clinicians. By addressing these disparities and striving for collaboration, the healthcare community can work towards achieving improved health outcomes for individuals living with SCI.
Minor corrections are needed. You find them noted in the pdf version of the article provided.
line 13: PLEX - explain the abbreviation the first time it appears in the text
line 15: (PLEX) - use only the abbreviation
line 43: British Columbia (BC) - abbreviation already explained above, please use only BC
line 84: punctuation
line 140: PI - explain the abbreviation
Figure 1: more clarity is needed for better visualization
line 413: extra punctuation

Reviewer 3 Report
Comments and Suggestions for Authors
Title: "Enhancing Clinician Confidence in Spinal Cord Injury Care: A Comprehensive Evaluation of Regional Workshops in British Columbia"
Strengths of the Article:
This paper addresses the challenges of accessing specialized spinal cord injury (SCI) care in the British Columbia (BC) Interior, where resources are limited. The focus is on identifying priorities and improvement needs in SCI health service delivery. An SCI-health community survey involved 44 individuals with lived experience (PLEX) in the BC Interior, highlighting 25 ranked topics. Top priorities included Bowel & Bladder Management (ranked 1), Sexual Health (5), and Pressure Injuries (PI) (7). In contrast, 102 clinicians in the BC Interior independently ranked 14 of these topics as lower clinical priorities (11-14).
These divergent perspectives informed a series of SCI clinical education workshops conducted in three regional cities. The aim was to align priorities and enhance clinicians' knowledge and confidence in managing SCI health. Feedback surveys from the workshops indicated that educational interventions, backed by lived experience perspectives (42%), effectively improved SCI health services and bridged the gap between clinician and PLEX viewpoints. However, it was emphasized that additional interventions, clinical training sessions, and resources are necessary to adequately address community needs and priorities.
Critiques and Suggestions:
The article acknowledges the journey for individuals with SCI, noting both promising medical advancements and existing challenges within a fragmented care system. Overall, disparities in priorities between PLEX and clinicians were observed. The project highlights the necessity and value of clinical and educational workshops presented by a combined clinical and PLEX team to improve clinician confidence and knowledge in providing SCI care. However, the paper requires some minor improvements to enhance clarity. Suggestions for refinement are detailed below.
The majority of clinicians who participated in the survey were predominantly physical and occupational therapists. It is important to highlight that these professionals might not naturally appreciate the significance of specific topics, such as bowel and bladder management, as these aspects typically lie beyond the usual focus of their clinical practice, as acknowledged in the limitations of the study. The decision to exclude other clinician groups from the survey is not explicitly explained, creating a need for clarification regarding why a more diverse range of clinicians was not included in the study.
Minor points:
1. Remove abbreviations from the Abstract.
2. Avoid abbreviations in the headings e.g.: 3.1.1. PLEX Community Priorities and others.
3. Some abbreviations were used only once like; line 335; “technology-enabled knowledge translation (TEKT)”. Please remove it from the text it is unnecessary.
Comments on the Quality of English LanguageThe English language is fine.
Round 2
Reviewer 1 Report
Comments and Suggestions for Authors
Thank you to the authors for their detailed responses to my previous comments. I have a few follow-up concerns that still need to be addressed detailed below:
· I still have concerns about the comparison of topics of interest between PLEX and clinicians. Specifically, 1) As stated in the responses, “[Ranked clinician priorities] was left to the interpretation of the clinicians who responded as the intent was to find out what they wanted to learn more about as clinicians.” This is very different from asking the priorities asked of the PLEX community, which is presumably asking what their personal healthcare priorities were. The vast majority of clinician attendees were O.T.’s/P.T.’s/rehabilitation assistants. Therefore, the most relevant information for their career would be spasticity/upper extremity function/pain/etc., and sexual health/bowel/bladder function would all likely fall outside of their interests. 2) I am still hung up on the list of priorities being different for the PLEX and clinician surveys. I understand that some items were not included because they were not included as part of the workshops, but the lack of similarity precludes a true comparison. Overall, the comparison between PLEX and clinician priorities makes up a large percentage of the abstract, results, and discussion, but the comparison is between fundamentally different questions and uses different sets of responses. This is a fundamental issue of the manuscript.
· The responses provided to each of the major comments sufficiently answered my other concerns. However, some responses did not result in changes to the manuscript. There are multiple responses that suggest that certain aspects of the paper are limitations or that will be addressed in future studies. However, these are not listed as limitations or as future directions in the text, which should be addressed.
· Minor: In Figure 1, the colored diagonal lines overlap the numbers on the right column causing some occlusion.
